# Greedy Sampling for Approximate Clustering in the Presence of Outliers

**Aditya Bhaskara**
University of Utah
bhaskaraaditya@gmail.com

**Sharvaree Vadgama**
University of Utah
sharvaree.vadgama@gmail.com

**Hong Xu**
University of Utah
hxu.hongxu@gmail.com

## Abstract

Greedy algorithms such as adaptive sampling ($k$-means++) and furthest point traversal are popular choices for clustering problems. One the one hand, they possess good theoretical approximation guarantees, and on the other, they are fast and easy to implement. However, one main issue with these algorithms is the sensitivity to noise/outliers in the data. In this work we show that for $k$-means and $k$-center clustering, simple modifications to the well-studied greedy algorithms result in nearly identical guarantees, while additionally being robust to outliers. For instance, in the case of $k$-means++, we show that a simple thresholding operation on the distances suffices to obtain an $O(\log k)$ approximation to the objective. We obtain similar results for the simpler $k$-center problem. Finally, we show experimentally that our algorithms are easy to implement and scale well. We also measure their ability to identify noisy points added to a dataset.

## 1 Introduction

Clustering is one of the fundamental problems in data analysis. There are several formulations that have been very successful in applications, including $k$-means, $k$-median, $k$-center, and various notions of hierarchical clustering (see [19, 12] and references there-in).

In this paper we will consider $k$-means and $k$-center clustering. These are both extremely well-studied. The classic algorithm of Gonzalez [16] for $k$-center clustering achieves a factor 2 approximation, and it is NP-hard to improve upon this for general metrics, unless P equals NP. For $k$-means, the classic algorithm is due to Lloyd [23], proposed over 35 years ago. Somewhat recently, [4] (see also [25]) proposed a popular variant, known as "$k$-means++". This algorithm remedies one of the main drawbacks of Lloyd's algorithm, which is the lack of theoretical guarantees. [4] proved that the $k$-means++ algorithm yields an $O(\log k)$ approximation to the $k$-means objective (and also improves performance in practice). By way of more complex algorithms, [21] gave a local search based algorithm that achieves a constant factor approximation. Recently, this has been improved by [2], which is the best known approximation algorithm for the problem. The best known hardness results rule out polynomial time approximation schemes [3, 11].

The algorithms of Gonzalez (also known as *furthest point traversal*) and [4] are appealing also due to their simplicity and efficiency. However, one main drawback in these algorithms is their sensitivity to corruptions/outliers in the data. Imagine $10k$ of the points of a dataset are corrupted and the coordinates take large values. Then both furthest point traversal as well as $k$-means++ end up choosing *only* the outliers. The goal of our work is to remedy this problem, and achieve the simplicity and scalability of these algorithms, while also being robust in a provable sense.

Specifically, our motivation will be to study clustering problems when some of the input points are (possibly adversarially) corrupted, or are outliers. Corruption of inputs is known to make even simple learning problems extremely difficult to deal with. For instance, learning linear classifiers in the presence of even a small fraction of noisy labels is a notoriously hard problem (see [18, 5]

and references therein). The field of high dimensional robust statistics has recently seen a lot of progress on various problems in both supervised and unsupervised learning (see [20, 14]). The main difference between our work and the works in robust statistics is that our focus is not to estimate a parameter related to a distribution, but to instead produce clusterings that are near-optimal in terms of an objective that is defined *solely on inliers*.

**Formulating clustering with outliers.** Let $\text{OPT}_{\text{full}}(X)$ denote the $k$-center or $k$-means objective on a set of points $X$. Now, given a set of points that also includes outliers, the goal in clustering with outliers (see [7, 17, 22]) is to partition the points $X$ into $X_{\text{in}}$ and $X_{\text{out}}$ so as to minimize $\text{OPT}_{\text{full}}(X_{\text{in}})$. To avoid the trivial case of setting $X_{\text{in}} = \emptyset$, we rquire $|X_{\text{out}}| \leq z$, for some parameter $z$ that is also given. Thus, we define the optimum $\text{OPT}$ of the $k$-clustering with outliers problem as

$$\text{OPT} := \min_{|X_{\text{out}}| \leq z} \text{OPT}_{\text{full}}(X \setminus X_{\text{out}}).$$

This way of defining the objective has also found use for other problems such as PCA with outliers (also known as robust PCA, see [6] and references therein). For the problems we consider, namely $k$-center and $k$-means, there are many existing works that provide approximation algorithms for $\text{OPT}$ as defined above. The early work of [7] studied the problem of $k$-median and facility location in this setup. The algorithms provided were based on linear programming relaxations, and were primarily motivated by the theoretical question of the power of such relaxations. Recently, [17] gives a more practical *local search* based algorithm, with running time quadratic in the number of points (which can also be reduced to a quadratic dependence on $z$, in the case $z \ll n$). Both of these algorithms are *bi-criteria* approximations (defined formally below). In other words, they allow the algorithm to discard $> z$ outliers, while obtaining a good approximation to the objective value $\text{OPT}$. In practice, this corresponds to declaring a small number of the *inliers* as outliers. In applications where the true clusters are robust to small perturbations, such algorithms are acceptable.

The recent result of [22] (and the earlier result of [10] for $k$-median) go beyond bi-criteria approximation. They prove that for $k$-means clustering, one can obtain a factor 50 approximation to the value of $\text{OPT}$, while declaring at most $z$ points as outliers, as desired. While this effectively settles the complexity of the problem, there are many key drawbacks. First, the algorithm is based on an iterative procedure that solves a linear programming relaxation in each step, which can be very inefficient in practice (and hard to implement). Further, in many applications, it may be necessary to improve on the (factor 50) approximation guarantee, potentially at the cost of choosing more clusters or slightly weakening the bound on the number of outliers.

Our main results aim to address this drawback. We prove that very simple variants of the classic Gonzalez algorithm for $k$-center, and the $k$-means++ algorithm for $k$-means result in approximation guarantees. The catch is that we only obtain bi-criteria results. To state our results, we will define the following notion.

**Definition 1.** *Consider an algorithm for the $k$-clustering (means/center) problem that on input $X, k, z$, outputs $k'$ centers (allowed to be slightly more than $k$), along with a partition $X = X'_{in} \cup X'_{out}$ that satisfies (a) $|X'_{out}| \leq \beta z$, and (b) the objective value of assigning the points $X'_{in}$ to the output centers is at most $\alpha \cdot \text{OPT}$.*

*Then we say that the algorithm obtains an $(\alpha, \beta)$ approximation using $k'$ centers, for the $k$-clustering problem with outliers.*

Note that while our main results only output $k$ centers, clustering algorithms are also well-studied when the number of clusters is not strictly specified. This is common in practice, where the application only demands a rough bound on the number of clusters. Indeed, the $k$-means++ algorithm is known to achieve much better approximations (constant as opposed to $O(\log k)$) for the problem without outliers, when the number of centers output is $O(k)$ instead of $k$ [1, 26].

## 1.1 Our results.

**K-center clustering in metric spaces.** For $k$-center, our algorithm is a variant of furthest point traversal, in which instead of selecting the furthest point from the current set of centers, we choose a *random point* that is not too far from the current set. Our results are the following.

**Theorem 1.1.** *Let $z, k, \varepsilon > 0$ be given parameters, and $X = X_{in} \cup X_{out}$ be a set of points in a metric space with $|X_{out}| \leq z$. There is an efficient randomized algorithm that with probability $\geq 3/4$ outputs a $(2 + \varepsilon, 4 \log k)$-approximation using precisely $k$ centers to the $k$-center with outliers problem.*

**Remark – guessing the optimum.** The additional $\varepsilon$ in the approximation is because we require guessing the value of the optimum. This is quite standard in clustering problems, and can be done by a binary search. If OPT is assumed to lie in the range $(c, c\Delta)$ for some $c > 0$, then it can be estimated up to an error of $c\varepsilon$ in time $O(\log(\Delta/\varepsilon))$, which gets added as a factor in the running time of the algorithm. In practice, this is often easy to achieve with $\Delta = \text{poly}(n)$. We will thus assume a knowledge of the optimum value in both our algorithms.

Also, note that the algorithm outputs exactly $k$ centers, and obtains the same (factor 2, up to $\varepsilon$) approximation to the objective as the Gonzalez algorithm, but after discarding $O(z \log k)$ points as outliers. Next, we will show that if we allow the algorithm to output $> k$ centers, one can achieve a better dependence on the number of points discarded.

**Theorem 1.2.** *Let $z, k, c, \varepsilon > 0$ be given parameters, and $X = X_{in} \cup X_{out}$ be a set of points in a metric space with $|X_{out}| \leq z$. There is an efficient randomized algorithm that with probability $\geq 3/4$ outputs a $(2 + \varepsilon, (1 + c)/c)$-approximation using $(1 + c)k$ centers to the $k$-center w/ outliers problem.*

As $c$ increases, note that the algorithm outputs very close to $z$ outliers. In other words, the number of points it *falsely discards* as outliers is small (at the expense of larger $k$).

**K-means clustering.** Here, our main contribution is to study an algorithm called T-kmeans++, a variant of $D^2$ sampling (i.e. $k$-means++), in which the distances are thresholded appropriately before probabilities are computed. For this simple variant, we will establish robust guarantees that nearly match the guarantees known for $k$-means++ without any outliers.

**Theorem 1.3.** *Let $z, k, \beta$ be given parameters, and $X = X_{in} \cup X_{out}$ be a set of points in Euclidean space with $|X_{out}| \leq z$. There is an efficient randomized algorithm that with probability $\geq 3/4$ gives an $(O(\log k), O(\log k))$-approximation using $k$ centers to the $k$-means with outliers problem on $X$.*

The algorithm outputs an $O(\log k)$ approximation to the objective value (similar to $k$-means++). However, the algorithm may discard up to $O(z \log k)$ points as outliers. Note also that when $z = 0$, we recover the usual $k$-means++ guarantee. As in the case of $k$-center, we ask if allowing a bi-criteria approximation improves the dependence on the number of outliers. Here, an additional dimension also comes into play. For $k$-means++, it is known that choosing $O(k)$ centers lets us approximate the $k$-means objective up to an $O(1)$ factor (see, for instance, [1, 4, 25]). We can thus ask if a similar result is possible in the presence of outliers. We show that the answer to both the questions is yes.

**Theorem 1.4.** *Let $z, k, \beta, c$ be given parameters, and $X = X_{in} \cup X_{out}$ be a set of points in a metric space with $|X_{out}| \leq z$. Let $\delta > 0$ be an arbitrary constant. There is an efficient randomized algorithm that with probability $\geq 3/4$ gives a $((\beta + 64), (1 + c)(1 + \beta)/c(1 - \delta))$-approximation using $(1 + c)k$ centers to the $k$-center with outliers problem on $X$.*

Given the simplicity of our procedure, it is essentially as fast as $k$-means++ (modulo the step of guessing the optimum value, which adds a logarithmic overhead). Assuming that this is $O(\log n)$, our running times are all $\widetilde{O}kn$. In particular, the procedure is significantly faster than local search approaches [17], as well as linear programming based algorithms [22, 10]. Our run times also compare well with those of recent, coreset based approaches to clustering with outliers, such as those of [9, 24] (see also references therein).

## 1.2 Overview of techniques

To show all our results, we consider simple randomized modifications of classic algorithms, specifically Gonzales' algorithm and the $k$-means++ algorithm. Our modifications, in effect, place a threshold on the probability of *any single point* being chosen. The choice of the threshold ensures that during the entire course of the algorithm, only a small number of outlier points will be chosen. Our analysis thus needs to keep track of (a) the number of points being chosen, (b) the number of *inlier* clusters from which we have chosen points (and in the case of $k$-means, points that are "close to the center"), (c) number of "wasted" iterations, due to choosing outliers. We use different *potential functions* to keep track of these quantities and measure progress. These potentials are directly inspired by the elegant analysis of the $k$-means++ algorithm provided in [13] (which is conceptually simpler than the original one in [4]).

## 2 Warm-up: Metric *k*-center in the presence of outliers

Let $(X, d)$ be a metric space. Recall that the classic Gonzalez algorithm [16] for $k$-center works by maintaining a set of centers $S$, and at each step finding the point $x \in X$ that is furthest from $S$ and adding it to $X$. After $k$ iterations, a simple argument shows that the $S$ obtained gives a factor 2 approximation to the best $k$ centers in terms of the $k$-center objective.

As we described earlier, this *furthest point traversal* algorithm is very susceptible to the presence of outliers. In particular, if the input $X$ includes $z > k$ points that are far away from the rest of the points, all the points selected (except possibly the first) will be outliers. Our main idea to overcome this problem is to ensure that no single point is too likely to be picked in each step. Consider the simple strategy of choosing one of the $2z$ points furthest away from $S$ (uniformly at random; we are assuming $n \geq 2z + k$). This ensures that in every step, there is at least a $1/2$ probability of picking an inlier (as there are only $z$ outliers). In what follows, we will improve upon this basic idea and show that it leads to a good approximation to the objective restricted to the inliers.

The algorithm for proving Theorems 1.1 and 1.2 is very simple: in every step, a center is added to the current solution by choosing a *uniformly random* point in the dataset that is at a distance $> 2r$ from the current centers. As discussed in Section 1.2, our proofs of both the theorems employ an appropriately designed potential function, adapted from [13].

---

**Algorithm 1** $k$-center with outliers

---

**Input:** points $X \subseteq \mathbb{R}^d$, parameters $k, z, r$; $r$ is a guess for OPT
**Output:** a set $S_\ell \subseteq X$ of size $\ell$
1: Initialize $S_0 = \emptyset$
2: **for** $t = 1$ to $\ell$ **do**
3:     Let $\mathcal{F}_t$ be the set of all points that are at a distance $> 2r$ from $S_{t-1}$. I.e.,

$$\mathcal{F}_t := \{x \in X : d(x, S_{t-1}) > 2r\}$$

4:     Let $x$ be a point sampled u.a.r from $\mathcal{F}_t$
5:     $S_t = S_{t-1} \cup \{x\}$
6: **return** $S_\ell$

---

**Notation.** Let $C_1, C_2, \ldots, C_k$ be the optimal clusters. So by definition, $\cup_i C_i = X_{\text{in}}$. Let $\mathcal{F}_t$ be the set of *far away* points at time $t$, as defined in the algorithm. Thus $\mathcal{F}_t$ includes both inliers and outliers. A simple observation about the algorithm is the following

**Observation 1.** *Suppose that the guess of $r$ is $\geq$ OPT, and consider any iteration $t$ of the algorithm. Let $u \in C_i$ be one of the chosen centers (i.e., $u \in S_t$). Then $C_i \cap \mathcal{F}_t = \emptyset$, and thus no other point in $C_i$ can be subsequently added as a center.*

Finally, we denote by $E_i^{(t)}$ the set of points in cluster $C_i$ that are at a distance $\geq 2r$ from $S_t$. I.e., we define $E_i^{(t)} := C_i \cap \mathcal{F}_t$. The observation above implies that $E_i^{(t)} = \emptyset$ whenever $S_t$ contains some $u \in C_i$. But the converse is not necessarily true (since all the points in $C_i$ could be at a distance $< 2r$ from points in other clusters, which happened to be picked in $S_t$).

Next, let $n_t$ denote the number of clusters $i$ such that $C_i \cap S_t = \emptyset$, i.e., the number of clusters none of whose points were selected so far. We are now ready to analyze the algorithm.

### 2.1 Algorithm choosing *k*-centers

We will now analyze the execution of Algorithm 1 for $k$ iterations, thereby establishing Theorem 1.1.

The key step is to define the appropriate potential function. To this end, let $w_t$ denote the number of times that one of the outliers was added to the set $S$ in the first $t$ iterations. I.e., $w_t = |X_{\text{out}} \cap S_t|$. The potential we consider is now:

$$\Psi_t := \frac{w_t |\mathcal{F}_t \cap X_{\text{in}}|}{n_t}. \tag{1}$$

Our main lemma bounds the expected increase in $\Psi_t$, conditioned on any choice of $S_t$ (recall that $S_t$ determines $n_t$).

**Lemma 1.** *Let $S_t$ be any set of centers chosen in the first $t$ iterations, for some $t \geq 0$. We have*

$$\mathbb{E}_{t+1}\left[\Psi_{t+1} - \Psi_t \mid S_t\right] \leq \frac{z}{n_t}.$$

As usual, $\mathbb{E}_{t+1}$ denotes an expectation only over the $(t+1)$th step. Let us first see how the lemma implies Theorem 1.1.

*Proof of Theorem 1.1.* The idea is to repeatedly apply Lemma 1. Since we do not know the values of $n_t$, we use the simple lower bound $n_t \geq k - t$, for any $t < k$.

Along with the observation that $\Psi_0 = 0$ (since $w_0 = 0$), we have

$$\mathbb{E}[\Psi_k] = \sum_{t=0}^{k-1} \mathbb{E}[\Psi_{t+1} - \Psi_t] \leq \sum_{t=0}^{k-1} \frac{z}{k-t} \leq zH_k,$$

where $H_k$ is the $k$th Harmonic number. Thus by Markov's inequality, $\Pr[\Psi_k \leq 4zH_k] \geq 3/4$. By the definition of $\Psi_k$, this means that with probability at least $3/4$,

$$\frac{w_k|\mathcal{F}_t \cap X_{\text{in}}|}{n_k} \leq 4z \ln k.$$

The key observation is that we always have $w_k = n_k$. This is because if the set $S_k$ did not intersect $n_k$ of the optimal clusters, then since $S_k$ cannot include two points from the same cluster (as we observed earlier), precisely $n_k$ of the iterations must have chosen outliers. This means that with probability at least $3/4$, we have $|\mathcal{F}_t \cap X_{\text{in}}| \leq 4z \ln k$. This means that after $k$ iterations, with probability at least $3/4$, at most $4z \ln k$ of the inliers are at a distance $> 2r$ away from the chosen set $S_k$. Thus the total number of points at a distance $> 2r$ away from $S_k$ is at most $z(4 \ln k + 1)$. This completes the proof of the theorem. $\qquad\square$

We thus only need to show Lemma 1.

*Proof of Lemma 1.* For simplicity, let us write $e_i := |E_i^{(t)}| = |C_i \cap \mathcal{F}_t|$. In other words $e_i$ is the number of points in the $i$th optimal cluster that are at distance $> 2r$ from $S_t$. Let us also write $F = \sum_i e_i$. By definition, we have that $F = |\mathcal{F}_t \cap X_{\text{in}}|$.

Then, the sampling in the $(t+1)$th iteration samples an inlier with probability $F/|\mathcal{F}_t|$, and an outlier with probability $1 - \frac{F}{|\mathcal{F}_t|}$. If an inlier is sampled, the value $n_t$ reduces by 1, but $w_t$ stays the same. If an outlier is sampled, the value $n_t$ stays the same, while $w_t$ increases by 1. The value of $|\mathcal{F}_t \cap X_{\text{in}}|$ is non-increasing. If a point in $C_i$ is chosen (which happens with probability $e_i/|\mathcal{F}_t|$), it reduces by at least $e_i$. Thus, we have

$$\mathbb{E}[\Psi_{t+1}] \leq \sum_{i=1}^{k} \frac{e_i}{|\mathcal{F}_t|} \frac{w_t(F - e_i)}{n_t - 1} + \left(1 - \frac{F}{|\mathcal{F}_t|}\right) \frac{(w_t + 1)F}{n_t}. \tag{2}$$

The first term on the RHS can be simplified as

$$\frac{w_t}{|\mathcal{F}_t|(n_t - 1)} \sum_i e_i(F - e_i) = \frac{w_t}{|\mathcal{F}_t|(n_t - 1)} \left(F^2 - \sum_i e_i^2\right)$$

The number of non-zero $e_i$ is at most $n_t$, by definition. Thus we have $\sum_i e_i^2 \geq F^2/n_t$. Plugging this into (2) and simplifying, we have

$$\mathbb{E}[\Psi_{t+1}] \leq \frac{w_t F^2}{|\mathcal{F}_t|n_t} + \left(1 - \frac{F}{|\mathcal{F}_t|}\right) \frac{(w_t + 1)F}{n_t} = \Psi_t + \left(1 - \frac{F}{|\mathcal{F}_t|}\right) \frac{F}{n_t}.$$

The proof now follows by using the simple facts: $\left(1 - \frac{F}{|\mathcal{F}_t|}\right) \leq \frac{z}{|\mathcal{F}_t|}$ (which is true because there are at most $z$ outliers) and $F \leq |\mathcal{F}_t|$ (which is true by definition, because $F = |X_{\text{in}} \cap \mathcal{F}_t|$). $\qquad\square$

This completes the analysis of Algorithm 1 when the number of centers $\ell$ is exactly $k$.

## 2.2 Bi-criteria approximation

Next, we see that running Algorithm 1 for $\ell = (1 + c)k$ iterations results in *covering* more clusters (thus resulting in fewer outliers). Thus we end up with a tradeoff between the number of centers chosen and the number of points the algorithm declares as outliers (while obtaining the same approximation (factor 2) for the objective OPT – Theorem 1.2). The potential function now needs modification. The details are deferred to Section A.1.

## 3 $k$-means via thresholded adaptive sampling

Next we consider the $k$-means problem when some of the points are outliers. Here we propose a variant of the $k$-means++ procedure (see [4]), which we call T-kmeans++. Our algorithm, like $k$-means++, is an iterative algorithm that samples a point to be a centroid at each iteration according to a probability that depends on the distance to the current set of centers. However, we avoid the problem of picking too many outliers by simply thresholding the distances.

**Notation.** Let us start with some notation that we use for the remainder of the paper. The points $X$ are now in a Euclidean space (as opposed to an arbitrary metric space in Section 2). We assume as before that $|X| = n$, and $X = X_{\text{in}} \cup X_{\text{out}}$, where $|X_{\text{out}}| = z$, which is a known parameter. Additioanlly, $\beta$ will be a parameter that we will control. For the purposes of defining the algorithm, we assume that we have a guess for the optimum objective value, denoted OPT.

Now, for any set of centers $C$, we define

$$\tau(x, C) = \min \left( d(x, C)^2, \frac{\beta \cdot \text{OPT}}{z} \right) \tag{3}$$

We follow the standard practice of defining the distance to an empty set to be $\infty$. Next, for any set of points $U$, define $\tau(U, C) = \sum_{x \in U} \tau(x, C)$. Note that the parameter $\beta$ lets us interpolate between uniform sampling ($\beta \to 0$), and classic $D^2$ sampling ($\beta \to \infty$). In our results, choosing a higher $\beta$ has the effect of reducing the number of points we declare as outliers, at the expense of having a worse guarantee on the approximation ratio for the objective.

We can now state our algorithm (denoted Algorithm 2)

---

**Algorithm 2** Thresholded Adaptive Sampling – T-kmeans++

**Input:** a set of points $X \subseteq \mathbb{R}^d$, parameters $k, z$, and a guess for the optimum OPT.
**Output:** a set $S \subseteq X$ of size $\ell$.
1: Initialize $S_0 = \emptyset$.
2: **for** $t = 1 \ldots \ell$ **do**
3:      sample a point $x$ from the distribution

$$p(x) = \frac{\tau(x, S_{t-1})}{\sum_{x \in X} \tau(x, S_{t-1})}. \qquad \text{(with } \tau \text{ as defined in (3))}$$

4:      set $S_t = S_{t-1} \cup \{x\}$.
5: **return** $S_\ell$

---

The key to the analysis is the following observation, that instead of the $k$-means objective, it suffices to bound the quantity $\sum_{x \in X} \tau(x, S_\ell)$.

**Lemma 2.** *Let $C$ be a set of centers, and suppose that $\tau(X, C) \leq \alpha \cdot$ OPT. Then we can partition $X$ into $X'_{in}$ and $X'_{out}$ such that*

1. $\sum_{x \in X'_{in}} d(x, C)^2 \leq \alpha \cdot$ OPT*, and*

2. $|X'_{out}| \leq \frac{\alpha z}{\beta}$.

*Proof.* The proof follows easily from the definition of $\tau$ (Eq. (3)). Let $X'_{\text{out}}$ be the set of points for which $d(x, C)^2 > \beta \text{OPT}/z$, and let $X'_{\text{in}}$ be $X \setminus X'_{\text{out}}$. Then by definition (and the bound on $\tau(X, C)$),

we have

$$\sum_{x \in X'_{\text{in}}} d(x, C)^2 + |X'_{\text{out}}| \frac{\beta \cdot \text{OPT}}{z} \leq \alpha \cdot \text{OPT}.$$

Both the terms on the LHS are non-negative. Using the fact that the first term is non-negative gives the first part of the lemma, and the inequality for the second term gives the second part of the lemma. □

### 3.1 $k$-means with outliers: an $O(\log k)$ approximation

Our first result is an analog of the theorem of [4], for the setting in which we have outliers in the data. As in the case of $k$-center clustering, we use a potential based analysis (inspired from [13]).

**Theorem 3.1.** *Running algorithm 2 for $k$ iterations outputs a set $S_k$ that satisfies*

$$\mathbb{E}[\tau(X, S_k)] \leq (\beta + O(1)) \log k \cdot \text{OPT}.$$

We note that Theorem 3.1 together with Lemma 2 directly implies Theorem 1.3. Thus the main step is to prove Theorem 3.1. This is done using a potential function as before, but requires a more careful argument than the one for $k$-center (specifically, the goal is not to include *some* point from a cluster, but to include a "central" one). Please see the supplement, section A.2 for details.

### 3.2 Bi-criteria approximation

**Theorem 3.2.** *Consider running Algorithm 2 for $\ell = (1 + c)k$ iterations, where $c > 0$ is a constant. Then for any $\delta > 0$, with probability $\geq \delta$, the set $S_\ell$ satisfies*

$$\tau(X, S_\ell) \leq \frac{(\beta + 64)(1 + c)\text{OPT}}{(1 - \delta)c}.$$

Note that this theorem directly implies Theorem 1.4 by repeating the algorithm $O(1/\delta)$ times. Once again, we use a slightly different potential function from the one for the $O(\log k)$ approximation. We defer the details of the proof to Section A.3 of the supplementary material.

## 4 Experiments

In this section, we demonstrate the empirical performance of our algorithm on multiple real and synthetic datasets, and compare it to existing heuristics. We observe that the algorithm generally behaves better than known heuristics, both in accuracy and (especially in) the running time. Our real and sythetic datasets are designed in a manner similar to [17]. All real datasets we use are available from the UCI repository [15].

$k$-**center with outliers.** We will evaluate Algorithm 1 on synthetic data sets, where points are generated according a mixture of $d$-dimensional Gaussians. The outliers in this case are chosen randomly in an appropriate bounding box.

**Metrics.** For $k$-center, we choose synthetic datasets because we wish to measure the *cluster recall*, i.e., the fraction of true clusters from which points are chosen by the algorithm. (Ideally, if we choose $k$ centers, we wish to have precisely one point chosen from each cluster, so the cluster recall is 1). We compute this quantity for three algorithms: the first is the trivial baseline of choosing $k'$ random points from the dataset (denoted **Random**). The second and third are **KC-Outlier** and **Gonzalez** respectively. Figure 1 shows the recall as we vary the number of centers chosen. Note that when $k = 20$, even when

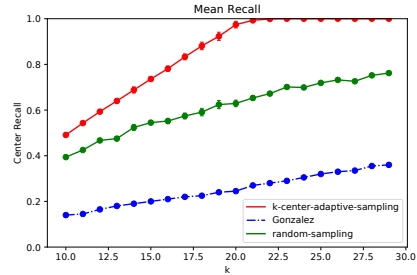

Figure 1: Figure showing cluster recall for the three algorithms, when $d = 15$, $k = 20$, $z = 100$ and $n = 10120$. The $x$ axis shows the number of clusters we pick.

roughly $k' = 23$ centers are chosen, we have a perfect recall (i.e., all the clusters are chosen) for our algorithm. Meanwhile **Random** and **Gonzalez** take considerably longer to find all the clusters.

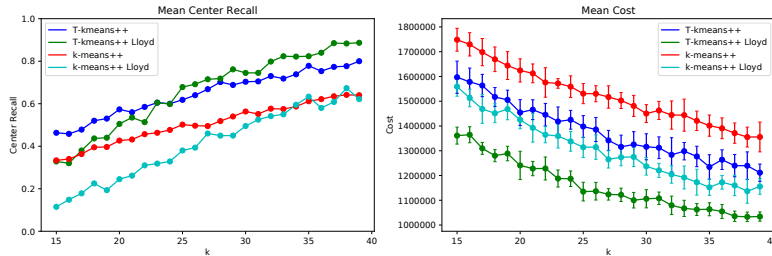

Figure 2: Figure showing the empirical cluster recall for the T-kmeans++ algorithm compared to prior heuristics. Here $k = 20, z = 2000, n = 12020$. The $x$ axis shows the number of clusters we pick.

$k$**-means with outliers.** Once again, we demonstrate the *cluster recall* on a synthetic dataset. In this case, we compare our algorithm with a heuristic proposed in [17]: running $k$-means++ followed by an iteration of "outlier-senstive Lloyd's iteration", proposed in [8]. We also ran the latter procedure as a post-processing step for our algorithm. Figure 2 reports the cluster recall and the value of the $k$-means objective for the algorithms. Unlike the case of $k$-center, the T-kmeans++ algorithm can potentially choose points in one cluster multiple times. However, we consistently observe that T-kmeans++ outperforms the other heuristics.

Finally, we perform experiments on three datasets:

1. NIPS (a dataset from the conference NIPS over 1987-2015): clustering was done on the rows of a $11463 \times 50$ matrix (the number of columns was reduced via SVD).

2. The MNIST digit-recognition dataset: clustering was performed on the rows of a $60000 \times 40$ (again, SVD was used to reduce the number of columns).

3. Skin Dataset (available via the UCI database): clustering was performed on the rows of a $245,057 \times 3$ matrix (original dataset).

In order to simulate corruptions, we randomly choose $2.5\%$ of the points in the datasets and corrupt all the coordinates by adding independent noise in a pre-defined range. The following table outlines the results. We report the *outlier recall*, i.e., the number of true outliers designated as outliers by the algorithm. For fair comparison, we make all the algorithms output precisely $z$ outliers. Our results indicate slightly better recall values for T-kmeans++. For some data sets (e.g. Skin), the $k$-means objective value is worse for T-kmeans++. Thus in this case, the outliers are not "sufficiently corrupting" the original clustering.[1]

| Dataset | $k$ | KM recall | TKM recall | KM objective | TKM objective |
|---------|-----|-----------|------------|--------------|---------------|
| NIPS | 10 | 0.960 | 0.977 | 4173211 | 4167724 |
| | 20 | 0.939 | 0.973 | 4046443 | 4112852 |
| | 30 | 0.924 | 0.978 | 3956768 | 4115889 |
| Skin | 10 | 0.619 | 0.667 | 7726552 | 7439527 |
| | 20 | 0.642 | 0.690 | 5936156 | 5637427 |
| | 30 | 0.630 | 0.690 | 5164635 | 4853001 |
| MNIST | 10 | 0.985 | 0.988 | $1.546 \times 10^8$ | $1.513 \times 10^8$ |
| | 20 | 0.982 | 0.989 | $1.475 \times 10^8$ | $1.449 \times 10^8$ |
| | 30 | 0.977 | 0.986 | $1.429 \times 10^8$ | $1.412 \times 10^8$ |

Table showing outlier recall for KM ($k$-means++) and TKM (T-kmeans++) along with the $k$-means cost.

## 5    Conclusion

We proposed simple variants of known greedy heuristics for two popular clustering settings ($k$-center and $k$-means clustering) in order to deal with outliers/noise in the data. We proved approximation guarantees, comparing to the corresponding objectives on *only the inliers*. The algorithms are also easy to implement, run in $\widetilde{O}(kn)$ time, and perform well on both real and synthetic datasets.

## Footnotes

[1]An anonymous reviewer suggested experiments on the kddcup-1999 dataset (as in [9]). However, we observed that treating certain labels as outliers as done in the prior work is not meaningful: the outliers turn out to be closer to one of the cluster centers than many points in that cluster.

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
