[Supplementary Material]

## A Supplementary Material

### A.1 Bi-criteria approximation for $k$-center

For the analysis, and subsequently the proof of Theorem 1.2, we use a slightly different potential function. The potential from (1) is tricky to control when $\ell > k$, as $n_t$ can become 0. Also, we will use $|\mathcal{F}_t|$ instead of $|\mathcal{F}_t \cap X_{\text{in}}|$. The potential is thus simply

$$\Psi_t := w_t |\mathcal{F}_t|. \tag{4}$$

As before, the following lemma bounds the increase in the potential, conditioned on the chosen centers $S_t$.

**Lemma 3.** *For any $t \geq 0$ and any $S_t \subseteq X$,*

$$\mathbb{E}_{t+1}[\Psi_{t+1} - \Psi_t \mid S_t] \leq z.$$

*Proof.* The proof is along the lines of that of Lemma 1. As before, define $E_i^{(t)} = |C_i \cap \mathcal{F}_t|$, and let $e_i = |E_i^{(t)}|$ and write $F = \sum_i e_i$. Once again, if the point chosen in the $(t+1)$th iteration is from $C_i$, then the quantity $|\mathcal{F}_t|$ reduces by at least $e_i$. Thus we have

$$
\begin{aligned}
\mathbb{E}_{t+1}[\Psi_{t+1}] &\leq \sum_i \frac{e_i}{|\mathcal{F}_t|} w_t(|\mathcal{F}_t| - e_i) + \left(1 - \frac{F}{|\mathcal{F}_t|}\right)(w_t + 1)|\mathcal{F}_t| \\
&= \Psi_t - \frac{w_t}{|\mathcal{F}_t|} \sum_i e_i^2 + \left(1 - \frac{F}{|\mathcal{F}_t|}\right)|\mathcal{F}_t| \qquad \left(\text{using } \sum_i e_i = F\right) \\
&\leq \Psi_t + \left(1 - \frac{F}{|\mathcal{F}_t|}\right)|\mathcal{F}_t| \leq \Psi_t + z. \tag{5}
\end{aligned}
$$

The first equality is obtained by rearranging the terms appropriately. In the last step, we used $|\mathcal{F}_t| - F \leq z$, as before. This completes the proof of the lemma. $\square$

We can now complete the proof of Theorem 1.2.

*Proof of Theorem 1.2.* Consider running the algorithm for $\ell = k(1+c)$ steps. By a repeated application of Lemma 3, we have that

$$\mathbb{E}[\Psi_\ell] \leq k(1+c)z.$$

Thus by Markov's inequality, we have that for any $\delta > 0$, the probability of the event $\Psi_\ell \leq \frac{k(1+c)z}{(1-\delta)}$ is at least $\delta$. Next, using the definition of $\Psi_\ell$, we have that with probability at least $\delta$,

$$w_\ell |\mathcal{F}_\ell| \leq \frac{k(1+c)z}{(1-\delta)}.$$

Now, if the algorithm is run for $k(1+c)$ iterations, at least $kc$ iterations are "wasted" (because once we pick a point from a cluster, the rest of the points get removed from $\mathcal{F}_t$). Thus we have $w_\ell \geq kc$. Thus with probability at least $\delta$, we have $|\mathcal{F}_\ell| \leq \frac{k(1+c)}{c(1-\delta)}$.

Thus given any $\delta > 0$, we can repeat the algorithm $O(1/\delta)$ times, and with high probability (at least $3/4$, say) one of the trials results in $|\mathcal{F}_\ell| \leq \frac{k(1+c)}{c(1-\delta)}$. This completes the proof of the theorem. $\square$

### A.2 Logarithmic approximation for $k$-means

In this section, we focus on proving Theorem 3.1.

**Notation.** In the remainder of the proof, we denote $\phi(x, S) = d(x, S)^2$, and $\phi(U, S) = \sum_{u \in U} d(u, S)^2$, for any $U, S \subseteq X$. We will also let $C_1, C_2, \ldots, C_k$ denote the optimal clusters. Thus we have $X_{\text{in}} = \cup_i C_i$.

The following is the so-called "parallel-axis theorem" (see, e.g., [4]).

**Proposition 1.** *Let $C \subset \mathbb{R}^d$ and let $\mu = \frac{1}{|C|} \sum_{x \in C} x$. Let $p$ be an arbitrary point in $\mathbb{R}^d$. Then*

$$\phi(C, \{p\}) = \phi(C, \{\mu\}) + |C| \cdot \|p - \mu\|^2.$$

The next two lemmas are taken from [4].

**Lemma 4** (Lemma 3.2 from [4]). *Let $C \subset \mathbb{R}^d$ be any set of points with mean $\mu$. Let $x$ be a point chosen uniformly at random from $C$. Then*

$$\mathbb{E}[\phi(C, \{x\})] = 2\phi(C, \{\mu\}) \tag{6}$$

The next lemma shows that if instead of the uniform distribution over $C$ (in Lemma 4), we choose each $x \in C$ with probability proportional to $\phi(x, T)$ for any set $T$, a similar inequality holds.

**Lemma 5** (Lemma 3.3 from [4]). *Let $C \subset \mathbb{R}^d$ be a set of points with mean $\mu$, and let $T \subseteq \mathbb{R}^d$ be another arbitrary set. Then we have*

$$\sum_{x \in C} \frac{\phi(x, T)}{\phi(C, T)} \cdot \phi(C, T \cup \{x\}) \leq 8\phi(C, \{\mu\}). \tag{7}$$

The main technical ingredient of our proof is proving that a similar inequality holds if points $x \in C$ are sampled proportional to $\tau(x, T)$ instead of $\phi(x, T)$.

**Lemma 6.** *Let $C \subset \mathbb{R}^d$ be a set of points with mean $\mu$, and let $T \subseteq \mathbb{R}^d$ be another arbitrary set. Then we have*

$$\sum_{x \in C} \frac{\tau(x, T)}{\tau(C, T)} \cdot \phi(C, T \cup \{x\}) \leq 64\phi(C, \{\mu\}). \tag{8}$$

For convenience, let us write $\Theta = \beta \text{OPT}/z$. We also denote $\phi^*(C) := \phi(C, \{\mu\})$.

To prove the lemma, we first show the following about the values $\{d(x, T)^2\}_{x \in C}$. This lemma will assume that $\phi(C, T) \geq 64\phi^*(C)$ (else Lemma 6 is trivial).

**Lemma 7.** *Suppose $\phi(C, T) \geq 64\phi^*(C)$. Then we have the following:*

*1. $\phi(C, T) \leq \frac{64}{31}|C|d(\mu, T)^2$.*

*2. Let $S \subseteq C$ be defined as $\{x \in C : d(x, T)^2 \in \left[\frac{1}{3}d(\mu, T)^2, \frac{7}{3}d(\mu, T)^2\right]\}$. Then we have $|S| \geq \frac{25}{31}|C|$.*

Roughly speaking, the lemma says that $d(x, T)^2$ values are *fairly uniform*, i.e., many of the values are close to $d(\mu, T)^2$.

*Proof.* We start by noting that by the triangle inequality (i.e., $\|x - y\|^2 \leq 2(\|x - z\|^2 + \|z - y\|^2)$), we have for any $x \in C$,

$$\frac{1}{2}d(\mu, T)^2 - \|x - \mu\|^2 \leq d(x, T)^2 \leq 2\left(\|x - \mu\|^2 + d(\mu, T)^2\right). \tag{9}$$

Summing the inequality on the right over all $x \in C$, we have

$$\phi(C, T) \leq 2|C|d(\mu, T)^2 + 2\phi^*(C).$$

Using the assumption that $\phi^*(C) \leq \phi(C, T)/64$ and simplifying, we get the first part of the lemma. For for the second part, define

$$S' = \{x \in C : \|x - \mu\|^2 \leq \frac{1}{6}d(\mu, T)^2\}.$$

By (9), we have that $S' \subseteq S$. Thus it suffices to lower bound $|S'|$. To do this, note that by Markov's inequality (since the sum of $\|x - \mu\|^2$ is $\phi^*$), we have

$$|C \setminus S'| \leq \frac{6\phi^*}{d(\mu, T)^2}.$$

Using the first part of the lemma (together with the lower bound on $d(C, T)$), we have that $d(\mu, T)^2 \geq 31\phi^*(C)/|C|$. Plugging this into the above, we have

$$|C \setminus S'| \leq \frac{6|C|}{31} \implies |S'| \geq \frac{25}{31}|C|.$$

This completes the proof of the lemma. $\qquad\square$

We are now ready to prove Lemma 6.

*Proof of Lemma 6.* We consider two cases. First, suppose $\Theta \geq \frac{7}{3}d(\mu, T)^2$. In this case, for all $x \in S$ (as defined in the statement of Lemma 7), we have $\tau(x, T) = d(x, T)^2 \geq d(\mu, T)/3$. Thus,

$$\tau(C, T) \geq |S| \cdot \frac{d(\mu, T)^2}{3} \geq \frac{25}{31}|C| \cdot \frac{31}{64} \frac{\phi(C, T)}{|C|} \cdot \frac{1}{3} \geq \frac{\phi(C, T)}{8}.$$

This implies that for *all* $x \in C$,

$$\frac{\tau(x, T)}{\tau(C, T)} \leq 8 \frac{\phi(x, T)}{\phi(C, T)}.$$

Thus, we can appeal to (7) to conclude the proof of Lemma 6 in this case.

Next, consider the case $\Theta < \frac{7}{3}d(\mu, T)^2$. In this case, for all $x \in S$, we have $\tau(x, T) = \min(\Theta, d(x, T)^2) \geq \Theta/7$. This implies that

$$\tau(C, T) \geq |S|\frac{\Theta}{7} \geq \frac{25}{31} \cdot \frac{1}{7} \cdot |C|\Theta \geq \frac{|C|\Theta}{10}.$$

Now by definition, we have $\tau(x, T) \leq \Theta$, and thus for all $x \in C$, we have

$$\frac{\tau(x, T)}{\tau(C, T)} \leq \frac{10}{|C|}.$$

Thus, we can now appeal to (6) to conclude the proof of the lemma. $\qquad\square$

The lemma immediately implies the following.

**Corollary 1.** *Consider step $t$ in the execution of Algorithm 2. Let $x$ be the the point chosen at the $t$'th step. Let $C$ be one of the optimal clusters, and let $\phi^*(C)$ be the contribution of the points in $C$ to the optimal cost. Then we have*

$$\mathbb{E}\left[\phi(C, S_{t-1} \cup \{x\}) \mid x \in C\right] \leq 64 \cdot \phi^*(C), \tag{10}$$
$$\mathbb{E}\left[\tau(C, S_{t-1} \cup \{x\}) \mid x \in C\right] \leq 64 \cdot \phi^*(C). \tag{11}$$

*Proof.* The proof of (10) follows from Lemma 6, using the fact that $\phi^*(C) = \phi(C, \{\mu\})$ in the case of an optimal cluster $C$. Eq. (11) follows from $\tau(C, S) \leq \phi(C, S)$ for any sets $C, S$. $\qquad\square$

We are now ready to prove Theorem 3.1. We will define a potential function as before. Consider the execution of the algorithm. We say that an optimal cluster $C_i$ is *covered* at time step $t$ if $C_i \cap S_t \neq \emptyset$. The number of *wasted iterations* $w_t$ until time $t$ is the number of iterations in which no new cluster is covered (this could be due to picking a point in an already-covered cluster, or due to picking an outlier). We also denote by $n_t$ the number of uncovered optimal clusters at time $t$. We let $H_t$ denote the union of points in covered (optimal) clusters, and $U_t$ be the union of points in uncovered (optimal) clusters (note that this does not include the outliers). In this notation, define the potential

$$\Psi_t = \frac{w_t \cdot \tau(U_t, S_t)}{n_t}.$$

As before, we will bound the expected increase in the potential $\Psi_{t+1} - \Psi_t$, conditioned on $S_t$.

**Lemma 8.** *Let $S_t$ be the set of points chosen in the first $t$ steps of the algorithm, and consider step $(t+1)$. We have*

$$\mathbb{E}[\Psi_{t+1} - \Psi_t \mid S_t] \leq \frac{\beta \cdot \mathrm{OPT} + \tau(H_t, S_t)}{n_t} \leq \frac{\beta \cdot \mathrm{OPT} + \tau(H_t, S_t)}{k - t}. \tag{12}$$

Before proving the lemma, let us see why it implies our theorem. We need another observation.

**Lemma 9.** *For any $t > 0$, we have*

$$\mathbb{E}[\tau(H_t, S_t)] \leq 64 \cdot \mathrm{OPT}. \tag{13}$$

*Proof.* Note that the expectation in (13) is over $S_t$. The lemma is then a direct consequence of Lemma 6. A formal proof of this can be shown via an inductive argument. Let $H'_t$ be the set of *indices* of the covered clusters (recall that $H_t$ is the union of the points in these clusters). Then we claim that for any $J \subseteq [k]$ of size $\leq t$,

$$\mathbb{E}[\tau(H_t, S_t) \mid H'_t = J] \leq 64 \sum_{j \in J} \phi^*(C_j).$$

This claim implies the lemma, by taking an expectation over $J$. The claim itself follows easily by induction, because we can expand the expectation on the LHS using all the choices for $H'_{t-1}$. Either no new cluster is covered in step $t$ (in which case $H_t = H_{t-1}$, and we can use the fact that $\tau(H_t, S_t) \leq \tau(H_t, S_{t-1})$), or a new cluster $j$ (for some $j \in J$) is covered in step $t$, in which case we can apply Lemma 6 along with the inductive hypothesis. $\square$

We can now complete the proof of Theorem 3.1.

*Proof of Theorem 3.1.* Combining Lemmas 8 and 9 and summing over $0 \leq t \leq k - 1$, we get that $\mathbb{E}[\Psi_k] \leq (\beta + 64) \log k \cdot \mathrm{OPT}$. $\square$

Thus it only remains to show Lemma 8.

*Proof of Lemma 8.* Conditioned on $S_t$, let us evaluate $\mathbb{E}[\Psi_{t+1}]$. Let $V$ denote the indices of the uncovered clusters (thus $|V| = n_t$). Then,

$$\mathbb{E}[\Psi_{t+1} \mid S_t] \leq \sum_{i \in V} \frac{\tau(C_i, S_t)}{\tau(X, S_t)} \frac{w_t \tau(U_t \setminus C_i, S_t)}{n_t - 1} + \frac{\tau(X \setminus U_t, S_t)}{\tau(X, S_t)} \frac{(w_t + 1)\tau(U_t, S_t)}{n_t}. \tag{14}$$

For convenience, write $\gamma_i = \tau(C_i, S_t)$, and let $\Gamma = \sum_{i \in V} \gamma_i$. Then (14) can be simplified as,

$$\mathbb{E}[\Psi_{t+1} \mid S_t] = \sum_{i \in V} \frac{w_t \gamma_i(\Gamma - \gamma_i)}{(n_t - 1)\tau(X, S_t)} + \left(1 - \frac{\Gamma}{\tau(X, S_t)}\right) \frac{(w_t + 1)\Gamma}{n_t}.$$

As in our analysis for $k$-center, we now use the fact that $\sum_{i \in V} \gamma_i(\Gamma - \gamma_i) \leq \Gamma^2(1 - \frac{1}{n_i})$. Plugging this in above and simplifying, we have

$$\mathbb{E}[\Psi_{t+1} \mid S_t] \leq \Psi_t + \left(1 - \frac{\Gamma}{\tau(X, S_t)}\right) \frac{\Gamma}{n_t}.$$

Now using the fact that $X \setminus U_t = H_t \cup X_{\mathrm{out}}$, along with the observation that $\tau(X_{\mathrm{out}}) \leq \beta \cdot \mathrm{OPT}$ (which follows from the definition of the threshold), the lemma follows. $\square$

### A.3 Bi-criteria guarantee for $k$-means with outliers

We now define a slightly different potential. We let $H_t, U_t$ be defined as before (Section 3.1).

$$\Phi_t := w_t \cdot \tau(X, S_t). \tag{15}$$

There are two differences here. First, we do not have a denominator of $n_t$. Second, we include $\tau(X, S_t)$ instead of $\tau$ restricted only to the uncovered inlier clusters. This makes the computation simpler, while also giving slightly better bounds.

467 **Lemma 10.** *Let $S_t$ be the points chosen in the first $t$ steps of the algorithm, and consider step $(t+1)$.*
468 *Then for $S_t$,*

$$\mathbb{E}[\Phi_{t+1} - \Phi_t \mid S_t] \le \beta\mathrm{OPT} + \tau(H_t, S_t).$$

469 Again, assuming Lemma 10, we can use Lemma 9 to conclude the proof of Theorem 3.2.

470 *Proof of Theorem 3.2.* By using Lemma 9 and summing over $t$, we have that

$$\mathbb{E}[\Phi_{(1+c)k}] \le (\beta + 64)\mathrm{OPT}(1 + c)k.$$

471 Thus by Markov's inequality, we have a probability at least $\delta$ of having

$$\Phi_{(1+c)k} \le \frac{(\beta + 64)(1 + c)k \cdot \mathrm{OPT}}{(1 - \delta)}.$$

472 Now, whenever we run for $(1 + c)k$ iterations, at least $ck$ of them have to be wasted (by definition,
473 there cannot be more than $k$ iterations in which a new cluster is covered). This implies that with
474 probability $\ge \delta$, the potential $\tau(X, S_\ell)$ satisfies the desired inequality.　□

475 We now turn to the proof of Lemma 10.

476 *Proof of Lemma 10.* The proof is actually simpler than the one for Lemma 8. We simply use the fact
477 that $\Phi(X, S_t)$ is monotonically decreasing with $t$ (because we only add elements to $S_t$). Thus,

$$\mathbb{E}[\Phi_{t+1} - \Phi_t] \le \Pr[w_{t+1} = w_t + 1] \cdot \tau(X, S_t).$$

478 I.e., the increase in potential is bounded by the probability that $w_t$ increases, times $\tau(X, S_t)$ (this
479 is true since $w_t$ increases by at most 1 in each iteration). The probability is precisely $\tau(X \setminus$
480 $U_t, S_t)/\tau(X, S_t)$ (i.e., the probability that we choose a point that is not in the uncovered clusters,
481 in other words, an outlier or an already covered point). Thus the probability is equal to $\tau(X_{\text{out}} \cup$
482 $H_t, S_t)/\tau(X, S_t)$. Plugging this into the equation above, we have

$$\mathbb{E}[\Phi_{t+1} - \Phi_t] \le \tau(X_{\text{out}}, S_t) + \tau(H_t, S_t) \le \beta\mathrm{OPT} + \tau(H_t, S_t).$$

483 This completes the proof.　□