[Reviews · NeurIPS 2019]

Reviewer 1



This paper proposes simple algorithmic modifications to popular approximation algorithms for the k-centers and k-means clustering problems. These modifications limit the selection of outliers, and thereby allow the authors to translate existing theoretical properties of the standard algorithms to the corresponding problems where outliers are present. The modifications in essence either constrain the selection of initial centroids based on their distance from the current set, or reduce the probabilities associated with selecting outliers which arise in the kmeans++ algorithm which has d^2 proportional probabilities. Empirical results show that the methods work well on some popular benchmarks. The paper is clear and well written, and the methods, although simple modifications of existing algorithms, are intuitive. The theoretical analysis is also well presented and persuasive. My main concern here surrounds the availability of a good estimate of OPT (the minimum objective value), especially for the k-means problem. The authors claim that this is a common assumption in clustering literature, but don't provide a reference. To conclude, a few minor comments/issue/typos: 1. Theorem 1.2 appears to dominate Theorem 1.1 when c = 1. If my understanding is correct, what then is the use of Theorem 1.1? 2. In Theorem 1.4 there is a typo in the statement of the approximation (extra right brace after "\beta + 64"). 3. In line 235 I presume you mean "... considerably more technical than our analysis for K-CENTERS..."

Reviewer 2



I have a mixed feeling about this paper. On one hand, it contains some nice and simple ideas such as threshold sampling (for k-means) and successive sampling (for k-center), which can potentially be useful in practice. On the other hand, I have the following concerns. First, it seems that the authors were not aware of recent works on clustering with outliers, including: [*] Distributed partial clustering, SPAA'17 [**] A practical algorithm for distributed clustering and outlier detection, NIPS'18 Though these algorithms are (mainly) designed for the distributed models, they can be used in the centralized setting as well. In fact, [**] used a successive sampling procedure (originally from "Optimal time bounds for approximate clustering", UAI'02) that is similar to Algorithm 1 in this paper. Certainly, the problems targeted in [**] are k-median/means, while Algorithm 1 is designed for k-center, but the underlying ideas (i.e., iterative sampling from uncovered points) look to be very similar. I hope the authors can make a careful comparison between these algorithms. Moreover, in [*] a centralized (O(1), 2)-bicriteria algorithm is designed for k-means with outliers. The algorithm uses k centers and has running time close to linear in terms of the dataset size. It looks like this result is strictly better than Thm 1.3 in terms of approximation guarantee? Second, the author mentioned in Section 3 that "Our first result is an analog of the theorem of [4], for the setting in which we have outliers in the data. As in the case of k-center clustering, we use a potential based analysis (inspired from [12])." I hope that some discussion on the novelty of the algorithm and analysis can be included in the main text. Otherwise it appears that Alg. 2 and its analysis are very incremental. The experiments part looks very brief. -- Please give the details about how the synthetic dataset is generated, at the level that others can repeat the experiments. -- The proposed algorithm should be compared with the one in [**], for k-means. -- There are real world data sets with ground truth (i.e., which are the outliers) available, such as KddCup99 (http://kdd.ics.uci.edu/databases/kddcup99/kddcup99.html). It will be good to see the performance of the proposed algorithm on real world datasets. -- Why for k-center one reports "cluster recall", while for k-means one reports "outlier recall"? It would be good to see both measurements on both cases. -- Better to give some intuition before the mathematical lemmas and proofs on why the extra threshold in the definition of \tau(x, C) helps in the outlier setting. Other comments: -- Intro, second paragraph, the last two sentences look to contradictory to each other? -- Alg. 1, why not directly use k instead of \ell?

Reviewer 3



- Experimental section does not compare the results of the suggested algorithms with the other known algorithms using other techniques such as local search. A more elaborate experimental section will help. - Even though the theoretical results are nice, the main selling point of the paper from a usability viewpoint is the simple algorithms that are fast and easy to implement (and hence debug). However, the running time analysis and comparison with other known methods are missing. Such an analysis will help see the results of this paper in the right perspective. - On the theoretical front, are there lower bounds arguments of the form: for fixed c=1 and a=2 is there an (a,b,c) algorithm with b = O(1)? There are many such combinations possible. Do the authors know about such lower bounds? It would be nice to include this in the discussion to be able to evaluate the nice upper bounds given in this work.

[Author Response · NeurIPS 2019]

We thank the reviewers for the detailed comments and suggested improvements.

**Reviewer 1: Estimate of OPT.** In all our algorithms, it suffices to know the optimum value up to a constant (say 2). Thus if we know a range for the value of OPT, one can perform a binary search. For instance, if we know that it lies in the interval $(1/n^{10}, n^{10})$ (a fairly large range), the search takes $O(\log n)$ time. Two early (arbitrarily chosen) examples of guessing the optimum in clustering problems are: *Clustering to Minimize the Sum of Cluster Diameters* (Charikar, Panigrahy, 2001), *A fast k-means implementation using coresets* (Frahling, Sohler, 2005). We will include more details about this step in the final version (possibly in the supplement).

**Reviewer 1: Typos.** We thank the reviewer for pointing these out. We will correct (2) and (3). As for (1), the number of centers needs to be $(1 + c)k$ as opposed to $ck$. We will correct this in the statements of theorems 1.2 and 1.4. This is why the theorems do not subsume theorems 1.1 and 1.3.

**Reviewer 2: Comparison to prior work.** We will compare and reference the works suggested by the reviewer. Indeed the works cited, as well as other "data reduction" approaches have been crucial to the development of algorithms for clustering. As our focus was on adaptive sampling approaches, we had not referred to those works earlier.

*Algorithm of (*) is better than Theorem 1.3:* This is indeed the case if "nearly linear time" is the main goal. However, note that the algorithm of (*) is based on iteratively reducing the size of the data, and is much more involved to describe. Meanwhile, our focus is to show that a simple variant of $k$-means++ itself achieves similar (though slightly worse guarantees). This is analogous to the case of vanilla (without outliers) $k$-means. Further, the bounds in Theorem 1.4 improve the approximation factor, albeit using more centers.

**Reviewer 2: Analysis of $k$-center vs $k$-means.** We will highlight at least some of the ideas involved in the $k$-means analysis in the body of the paper. The analysis is much more challenging because in $k$-means, it is no longer simply a matter of "covering" a cluster (i.e., choosing different points in a cluster lead to significantly different objective values for the other points). The lemmas in sections A.2 and A.3 of the supplement address this challenge.

**Reviewer 2: Running time analysis.** We will add this in the final version. The run time is $O(nk)$, the same as that for $k$-means++, as long as we have an estimate for the optimum value. Guessing that adds an extra logarithmic factor.

**Reviewer 2: Experiments.** In the final version (possibly in the supplement), we will add the details about the hyperparamters used in the synthetic experiments and in the noise addition step for real data. We will also perform experiments on the kdd-cup dataset (using only the numeric features and normalization as suggested in (**)).

**Reviewer 2: Other comments.** We will clarify the statements in the second paragraph of the introduction (latter line should say that a polynomial time approximation scheme is ruled out). The use of $\ell$ is because it is set to $(1 + c)k$ in the bi-criteria algorithms.

**Reviewer 3: Comparisons.** As discussed above, we will include more comparisons (in both running time and approximation factors $(a, b, c)$) with prior works. A short summary is as follows: if one is only concerned with *polynomial* running times, one can achieve $a = c = 1$ and $b = O(1)$ (Krishnaswamy, Li, Sandeep, STOC 2018). Using iterative "data reduction" approaches (cited above), one can achieve $c = 1$ while having $a = b = O(1)$, with the $O(1)$ term having a trade-off with the running time. Our algorithms (i) avoid such tradeoffs, and (ii) are simple modifications of well-studied greedy update procedures.

**Reviewer 3: Lower bounds.** The result of Krishnaswamy et al. (above) shows that $a = c = 1$ and $b = O(1)$ is indeed achievable. It is an interesting open question if the constant $b$ is worse for the outlier version of the problem. As for our algorithms, there are examples (based on the tight examples for $k$-means++) that indeed show that our *analysis* is tight. Thus improvements must come from more involved algorithms.

[Meta-Review · NeurIPS 2019]

The authors show that mild modifications to existing greedy iterative algorithms can make these robust to outliers. While the modifications are simple and looks incremental the reviewers feel that the analysis will be interesting to Neurips audience. The authors should cite the algorithms pointed out by the reviewers, since they are also based on successive sampling and can be thought of as adaptive methods. The experiments should include more comparisons as pointed out by the reviewers.